# Delivery of the VIVIT Peptide to Human Glioma Cells to Interfere with Calcineurin-NFAT Signaling

**DOI:** 10.3390/molecules26164785

**Published:** 2021-08-07

**Authors:** Aleksandra Ellert-Miklaszewska, Agata Szymczyk, Katarzyna Poleszak, Bozena Kaminska

**Affiliations:** Laboratory of Molecular Neurobiology, Nencki Institute of Experimental Biology, Polish Academy of Sciences, 02-093 Warsaw, Poland; agataszymczyk4@gmail.com (A.S.); k.poleszak@nencki.edu.pl (K.P.)

**Keywords:** NFAT proteins, calcineurin, interfering peptide, cell-penetrating peptide, glioma

## Abstract

The activation of NFAT (nuclear factor of activated T cells) transcription factors by calcium-dependent phosphatase calcineurin is a key step in controlling T cell activation and plays a vital role during carcinogenesis. NFATs are overexpressed in many cancers, including the most common primary brain tumor, gliomas. In the present study, we demonstrate the expression of NFATs and NFAT-driven transcription in several human glioma cells. We used a VIVIT peptide for interference in calcineurin binding to NFAT via a conserved PxIxIT motif. VIVIT was expressed as a fusion protein with a green fluorescent protein (VIVIT-GFP) or conjugated to cell-penetrating peptides (CPP), Sim-2 or 11R. We analyzed the NFAT expression, phosphorylation, subcellular localization and their transcriptional activity in cells treated with peptides. Overexpression of VIVIT-GFP decreased the NFAT-driven activity and inhibited the transcription of endogenous NFAT-target genes. These effects were not reproduced with synthetic peptides: Sim2-VIVIT did not show any activity, and 11R-VIVIT did not inhibit NFAT signaling in glioma cells. The presence of two calcineurin docking sites in NFATc3 might require dual-specificity blocking peptides. The cell-penetrating peptides Sim-2 or 11R linked to VIVIT did not improve its action making it unsuitable for evaluating NFAT dependent events in glioma cells with high expression of NFATc3.

## 1. Introduction

Short interfering peptides offer one of the most specific tools to target protein–protein interactions for research and potential therapeutic purposes. A properly designed peptide can mimic a particular binding motif and thereby precisely inhibit the interaction between two protein partners. However, poor membrane permeability of hydrophilic peptides prevents their spontaneous cell uptake and constitutes a severe limitation in the case of targeting intracellular signaling pathways. A promising solution to overcome this obstacle is the conjugation of an interfering peptide of interest with a cell-penetrating peptide (CPP), which has the capacity to ubiquitously cross cellular membranes with very limited toxicity, via energy-dependent and/or independent mechanisms, and without the necessity of recognition by specific receptors [1,2].

The NFAT (nuclear factor of activated T cells) transcription factors play a key regulatory role in the expression of genes coding for cytokines during activation and differentiation of T cells [3]. Pharmacological suppressors of NFAT signaling, such as tacrolimus (FK506) and cyclosporin A (CsA), are widely used as immunosuppressive drugs in the treatment of autoimmune diseases and transplant therapy. Four out of five NFAT family members, namely NFATc1 (NFAT2), NFATc2 (NFAT1), NFATc3 (NFAT4/NFATx), and NFATc4 (NFAT3), are activated by calcineurin, a calcium/calmodulin-dependent, serine/threonine-protein phosphatase. Responding to increased intracellular calcium levels, calcineurin docks to the regulatory domain of cytosolic NFAT and dephosphorylates multiple serine residues, which leads to the exposure of a nuclear localization sequence (NLS) of NFAT allowing its translocation to the nucleus and regulation of gene expression. Two binding sites for calcineurin on NFAT have been mapped: the major docking site—a PxIxIT motif, located near the N terminus of the regulatory region and an LxVP motif located near its C terminus. Optimization of the PxIxIT motif through affinity-driven selection from combinatorial peptide libraries led to the discovery of the VIVIT peptide (MAGPHPVIVITGPHEE), which shows higher binding affinity to calcineurin than the native NFATc2 docking sequence, PRIEIT [4]. This synthetic peptide selectively competes with NFAT for binding to calcineurin and prevents NFAT activation without affecting calcineurin phosphatase activity towards other protein substrates [4,5]. Further development led to the design of CPP-conjugated NFAT inhibitory peptides, which enabled translocation of the peptide through cellular membranes. Sim-2-VIVIT, which contains the Sim-2-CPP from the human Sim-2 transcription factor (AKAARQAARG), was used to inhibit T-cell activation and alleviate allergic airway inflammation [6]. Noguchi et al. [7] modified the VIVIT peptide at the N terminus with an 11-arginine transduction domain (11R) and a 3-glycine linker sequence. A synthetic 11R-VIVIT peptide (RRRRRRRRRRR-GGG-MAGPHPVIVITGPHEE) bearing polyarginine CPP could prevent the activation and proliferation of T cells both in vitro and in vivo, provided immunosuppression in a murine islet xenograft model [7], showed anti-inflammatory activity in lipopolysaccharide-stimulated macrophages and in experimental colitis in mice [8]; ameliorated diabetic nephropathy and podocyte injury in diabetic db/db mice [9]. Delivery of tat-VIVIT (with the HIV-1-derived peptide GRKKRRQRRRPQ) attenuated inflammatory reactions in the brain and deposition of amyloid Aβ plaques in a murine model of Alzheimer’s disease [10].

Besides their well-described role in the immune system, NFATs have a wide range of functions in other organs of the body. They are involved in the regulation of genes related to cell proliferation, differentiation, apoptosis, and angiogenesis, all of which may be crucial to tumor development [11]. NFATs are overexpressed in many types of cancers such as breast, pancreatic, lung, colorectal and hematological cancers [12]. Few studies, including ours, have demonstrated the expression and activity of NFAT factors in rat and human glioma cells [13,14,15,16,17,18]. The blocking of calcineurin/NFAT signaling using CsA led to glioma cell apoptosis and treatment with either CsA or FK506 decreased growth of mouse GL261 gliomas [19,20]. However, functions and transcriptional targets of NFAT proteins in glioma cells remain largely unknown. FK506 and CsA bind to intracellular peptidyl propyl isomerases (FKBP12 and cyclophilin A, respectively), and in such complexes inhibit calcineurin. Both drugs not only block calcineurin activity but also hinder the function of their endogenous targets, which limits their use as specific NFAT inhibitors. Therefore, we explored if the VIVIT peptide could be a tool for selective interference into calcienurin-driven activation of NFAT factors.

In the present study, we evaluated the expression/activity of NFAT proteins in glioma cells and the impact of inhibition of calcineurin/NFAT interaction by the VIVIT peptide. The scheme of the peptide delivery routes and the expected mode of action is presented in Appendix A. First, the VIVIT peptide was expressed as a fusion protein with a green fluorescent protein (GFP). Next, we compared the efficacy of two types of CPP, Sim-2 and 11R, in delivering the VIVIT peptide across the cellular membrane of glioma cells. We analyzed the phosphorylation and subcellular localization of NFATs, as well as their transcriptional activity toward a reporter luciferase gene and endogenous NFAT target genes.

## 2. Results

### 2.1. Human Glioma Cells Show Different Profiles of NFAT Expression and Activity

NFAT mRNA and protein levels were assessed in five established human glioma cell lines LN229, U87, U251, T98 and LN18, a primary glioma cell culture (WG4), and in normal human astrocytes (NHA). All glioma cell lines were derived from WHO grade III and IV brain tumors. mRNA levels of all NFAT family members (NFATc1–NFATc4) were evaluated by qPCR. All glioma cells showed high levels of *NFATc3*, while *NFATc1* and *NFATc2* expression levels were variable among the cells (Figure 1a). There was statistically significant upregulation of *NFATc2* and *NFATc3* in LN229 and U87 cells as compared to NHA. Notably, *NFATc1* mRNA was not detected in U87 cells. We found low expression of *NFATc4* in tested cells and due to this observation this factor was omitted in further studies. Western blot analysis confirmed the abundance of NFATc3 in tested glioma cells. NFATc1 was elevated in LN229 and T98 cells, NFATc2 showed detectable levels only in LN229 cells, while faint expression was visible in U87 and U251 cells (Figure 1b). The levels of all NFAT factors were very low in NHA. Differences in the pattern of bands in various cells on Western blots (especially NFATc1) result from altered electrophoretic mobility of NFATs due to the phosphorylation of multiple serine residues in the regulatory domain of NFATs.

In order to assess the transcriptional activity of NFAT factors, NHA and the glioma cells were transiently transfected with the pGL3-3xNFAT-luc plasmid, in which the reporter gene encoding luciferase is under the control of an NFAT-responsive promoter. This promoter contains a fragment of the *IL-2* gene promoter with 3 binding sites for NFAT proteins. The highest luciferase activity was detected in LN229 cells, which was in line with the abundance of all NFAT factors in these cells. U251 and LN18 cells, similarly to NHA, showed a low luciferase activity suggesting negligible NFAT signaling (Figure 1c). Since the basic activity of NFAT proteins in glioma cells may be relatively low, the cells were treated with a combination of 1 μM ionomycin (Io, a calcium ionophore) and 50 nM phorbol 12-myristate 13-acetate (PMA, an activator of protein kinase C). This resulted in an increase in intracellular calcium levels and activation of calcineurin, which led to increased activity of NFAT transcription factors and augmented luciferase activity. This also confirmed the specificity of the promoter activation by factors that are dependent on the calcium level in the cell. Based on these results we selected LN229, T98 and U87 cells for further experiments due to the highest expression and activity of NFATs in these cells.

### 2.2. VIVIT-GFP Blocks Transcriptional Activity of NFAT Factors

Activation of all NFATs, including NFATc3, can be blocked by the VIVIT peptide (MAGPHPVIVITGPHEE) [4]. This is due to a high sequence similarity in the major calcineurin binding motif, PxIxIT, shared by NFATc1–c3 (Figure 2a). In order to check if the VIVIT is active in glioma cells, we first overexpressed the VIVIT fused to GFP in LN229 cells using a construct coding for VIVIT-GFP. The efficacy of transfection reached over 70% and the cell viability was not affected (Figure 2b). In contrast to the mock-transfected cells and the cells transfected with a GFP-coding construct (CTRL, pEGFP-N1 vector), the cells overexpressing VIVIT-GFP showed an altered pattern of bands corresponding to NFATc1 as detected by Western blotting. A shift towards higher molecular weights suggested an appearance of highly phosphorylated, inactive NFATc1 (Figure 2c). Luciferase reporter assay using the pGL3-3xNFAT-luc plasmid showed that the overexpression of VIVIT-GFP in LN229 cells reduced the NFAT-dependent luciferase activity as compared to the cells transfected with CTRL construct (Figure 2d). Moreover, the cells transfected with VIVIT-GFP showed lower mRNA levels of the known endogenous NFAT-target genes, such as *COX2*, c-*MYC* and *CCND1*, relative to CTRL cells (Figure 2e).

### 2.3. Sim-2–Conjugated VIVIT Peptides Are Not Active in Glioma Cells

Next, we sought to determine whether the exogenous VIVIT peptides are capable of interfering with NFAT-calcineurin binding. To improve cell penetration, we used Sim2-CPP-modified peptides. LN229 and T98G cells were treated with 500 µM of one of the peptides: (i) Sim2-VIVIT, which contains the VIVIT and cell-penetrating Sim2 peptides; (ii) Sim2-VEET, which contains the same amino acid residues as the VIVIT but the last 12 amino acids were scrambled and the cell-penetrating Sim2 sequence, and (iii) VIVIT alone (Appendix A). The peptides did not show any effect on cell morphology and viability of LN229 and T98G cells, even at a very high concentration of 500 µM (not shown). As an additional control, the cells were treated with CsA, known to inhibit the calcineurin-dependent NFAT dephosphorylation. In both cell lines, CsA retarded the gel mobility of NFATc1 as a result of the accumulation of the hyperphosphorylated forms of NFATc1 (Figure 3a) and decreased luciferase activity in cells transfected with the pGL3-3xNFAT-luc plasmid (Figure 3b). None of the peptides mimicked the action of CsA on the pattern of NFATc1 bands in the Western blotting and did not affect the activity of NFAT-responsive promoter in the luciferase reporter assay (Figure 3a,b). 

### 2.4. Interfering Peptide Linked to Oligoarginine CPP Affects Calcineurin/NFAT Pathway 

As Sim2-conjugated peptides were not effective, we used the oligoarginine (11R) sequence as a CPP, linked to a VIVIT peptide (11R-VIVIT) or a control VEET peptide (11R-VEET) (Appendix A). During the initial screen using a peptide in a wide range of concentrations, 11R-VIVIT showed cytotoxicity in LN229 cells at and above 50 µM (not shown). To avoid harmful effects on cell viability, we treated the cells with the peptides at lower concentrations. The impact of the peptides on NFAT activity was first assessed using the luciferase reporter assay. The cells were pre-treated for 1 h with 2–20 µM 11R-VIVIT or a control 11R-VEET peptide before transfection with the pGL3-3xNFAT-luc plasmid. Unexpectedly, 11R-VIVIT at 10 µM and to a much greater extent at 20 µM caused activation of the NFAT-responsive promoter producing a significant increase in the luciferase activity as compared to untreated or 11R-VEET-treated cells (Figure 4a). Cell proliferation measured by BrdU incorporation assay was not affected by any of the peptides at those concentrations (Figure 4b). A similar set of experiments was repeated on U87 cells, which lack the expression of NFATc1. We found the upregulated luciferase activity upon 20 µM 11R-VIVIT as compared to untreated or 11R-VEET-treated cells but the increase was not statistically significant (Figure 4c). The proliferation of U87 cells was also not affected by the peptides (Figure 4d). Interestingly, in both LN229 and U87 cells, 11R-VIVIT upregulated the mRNA levels of two out of three tested endogenous NFAT-target genes, namely *COX2* and c-*MYC*, relative to untreated or 11R-VEET-treated cells. The increases in the gene expression were higher in LN229 than in U87 cells (Figure 4e,f).

We also checked whether the 11R-VIVIT treatment affected the *NFAT* mRNA levels that could explain the observed increase in NFAT signaling in LN229 cells. 11R-VIVIT upregulated *NFATc1* expression in a dose-dependent manner and, to a lesser extent, modulated *NFATc2* mRNA levels (Appendix A).

### 2.5. 11R-VIVIT Triggers Translocation of NFAT to the Nucleus

NFAT proteins upon dephosphorylation and activation change their intracellular localization. Blocking the interactions between calcineurin and NFAT using an interfering peptide should keep the phosphorylated factor in the cytoplasm and prevent its translocation to the nucleus. To investigate the effect of the 11R-VIVIT peptide on the intracellular localization of NFATs in glioma cells, LN229 and U87 cells were treated with 20 μM 11R-VIVIT or the control 11R-VEET peptide for 4 h and NFAT levels were evaluated in cytoplasmic and nuclear fractions using Western blotting (Figure 5a, Appendix A). CsA (5 µg/mL), which blocks calcineurin activity, leaves NFAT proteins inactive and in the cytoplasm. Cells treated with CsA constitute a reference control. Detection of GAPDH and Lamin B was used to assess the purity of the cytoplasmic and nuclear fractions, respectively, and to ensure loading of equal amounts of proteins. The levels of NFATc1–c3 decreased in the cytoplasmic fraction 4 h following administration of 11R-VIVIT in LN229 cells, which was accompanied by an increase in their levels in the nuclear fraction (Figure 5a,b). 11R-VEET-treated cells showed similar levels of NFATc1–c3 in both cellular fractions as the untreated cells. NFATc1 is undetectable in U87 cells, and low levels of NFATc2 in U87 cells hindered the detection of this protein after cellular fractionation. However, the accumulation of NFATc3 in the nuclear fraction of 11R-VIVIT-treated U87 cells is clearly visible (Appendix A). In both cell lines, CsA treatment resulted in a reduction in NFAT levels in the nuclear fraction, in accordance with its known mechanism of action.

Moreover, the effect of 11R-VIVIT on the intracellular localization of NFAT proteins was evaluated using immunocytochemical staining for NFATc1 and NFATc3, the most abundant NFATs in LN229 cells (Figure 5c,d). Two additional treatments were included: cells were treated with 5 µg/mL CsA (to inhibit the NFAT activity) or cells were stimulated with 1 μM Io and 50 nM PMA (these compounds activate NFAT). CsA blocked the dephosphorylation of NFATs preventing their translocation to the nucleus and thus most of the staining was detected in the cytoplasm. On the other hand, the action of calcineurin-NFAT pathway activators, Io and PMA, caused the accumulation of NFAT factors in the cell nuclei. The cells treated with the control peptide 11R-VEET showed a similar staining pattern as the untreated cells (CTRL). 11R-VIVIT induced translocation of NFATc1 and NFATc3 to the cell nucleus. 

## 3. Discussion

Since our first demonstration of the transcription factors NFAT expression and importance of calcienurin/NFAT pathway in rat glioma cells [13], several other reports confirmed the presence of calcineurin-dependent NFAT proteins in human glioma cells but processes regulated by these factors have not been fully understood [14,15,16,17]. In the present study, we demonstrate the expression of NFAT mRNAs and proteins in several human glioma cells and their lack in normal human astrocytes. To understand NFAT functions in glioma cells we evaluated the effects of the VIVIT peptide, which was designed as a high-affinity competitor for NFAT binding to calcineurin [4]. 

Both quantifications of *NFAT* expression by qPCR and Western blotting showed that NFATc3 is the most abundantly expressed member of this family in human glioma cells. LN229 glioma cells expressed all NFAT factors and presented the highest NFAT-dependent transcriptional activity. Studies by Wang and co-authors demonstrated the expression of NFATc1 in U251 glioblastoma cells and linked it to increased invasiveness and migration of cancer cells. NFATc1 acted via activation of *COX-2* expression, which is an inducer of invasion and migration in many cancer cells [15]. Moreover, NFATc1 was found constitutionally active in human glioblastoma samples [15]. Microarray analysis demonstrated that *NFATc2* was overexpressed in glioblastoma when compared to low-grade gliomas, and the expression of the NFATc2 protein in U87 and U251 glioblastoma cells has been linked to their increased invasiveness [14]. NFATc3 was the predominant factor detected in U251 cells and a collection of primary human glioblastoma cell lines [16], which corroborates our results. Knockdown of NFATc3 affected proliferation and migration of glioma cells in vitro and orthotropic tumor growth in mice [16].

Calcineurin inhibitors, such as tacrolimus (FK506) and CsA, which prevent activation of NFAT and NFAT-driven expression of immune response genes, have revolutionized transplant therapy. However, their molecular targets go beyond inhibition of NFAT signaling and their use is associated with adverse effects including progressive loss of renal function, cardio- and neurotoxicity, and increased risk of malignancy. Compounds that interfere selectively with the calcineurin-NFAT interaction without affecting its phosphatase activity may be useful as therapeutic agents that are less toxic than current drugs. We employed the VIVIT peptide [4], which competes with NFAT at calcineurin docking site. Our results show that forced expression of the construct encoding the VIVIT-GFP peptide prevents the binding of NFATs to calcineurin and their subsequent dephosphorylation, resulting in the reduced activity of the NFAT-driven promoter and inhibition of transcription of endogenous NFAT target genes. 

To deliver VIVIT to glioma cells, we used a Sim2 peptide, as the Sim2-conjugated VIVIT showed the inhibitory activity on NFAT signaling in other cells [6]. However, we failed to detect any effect of the Sim2-VIVIT on NFAT signaling in LN229 and T98 glioma cells. The pattern of NFATc1 bands detected by immunoblotting and the NFAT-dependent luciferase activity in the cells treated with Sim2-VIVIT were similar to those in untreated controls and in cells treated with the VIVIT peptide alone (without CPP, unable to enter the cell) or the scrambled Sim2-VEET peptide. Next, we tested the previously described 11R-VIVIT and 11R-VEET peptides [7]. Unexpectedly, 11R-VIVIT upregulated the transcriptional activity of the NFAT-driven luciferase promoter, as well as the expression of NFAT-regulated genes, such as *COX2* and *cMYC*. Moreover, 11R-VIVIT-treated cells showed increased expression of *NFATc1* and *NFATc2* genes. Although the elevated levels of these transcripts could explain to some extent the observed activation of NFAT signaling, posttranslational modifications affecting an intracellular localization of NFAT are the essential regulators of NFAT activity. We found that 11R-VIVIT, but not the scrambled peptide 11R-VEET, triggered a decrease in NFATc1-c3 in the cytoplasm and their increase in the nuclear fraction. We corroborated Western blot results by immunocytochemistry and demonstrated a nuclear import of NFAT factors upon 11R-VIVIT treatment. Thus, the response of glioma cells to 11R-VIVIT resembled the stimulatory effect of Io/PMA rather than the inhibitory action of CsA on NFAT signaling. Therefore, 11R-VIVIT is not suitable for inhibition of NFAT signaling and evaluating NFAT targets in glioma cells.

In the original study on the VIVIT peptide, Aramburu and co-authors demonstrated a blocking activity of VIVIT against NFATc1, NFATc2 and NFATc3 proteins [4]. However, these experiments were carried out on cell lysates after overexpression of each NFAT protein. The inhibitory efficacy of the extracellular 11R-VIVIT peptide has been tested in immune cells, where NFATc2 accounted for 90% of the NFAT transcriptional activity [7]. Other researchers effectively blocked NFAT proteins using the VIVIT peptide in endothelial cells and vascular smooth muscle cells [21], in which NFATc2 and NFATc1 predominate. Noteworthy, the calcineurin docking site of NFATc3 within the PxIxIT motif (CPSIQIT) varies from the sequence shared by NFATc1 and NFATc2 (SPRIEIT). This could contribute to different effects of 11R-VIVIT in glioma cells, which express mainly NFATc3. Moreover, apart from a PxIxIT motif localized to N-terminus, the C-terminus of NFAT contains another consensus motif, LxVP, that facilitates calcineurin docking and NFAT dephosphorylation [22]. Interestingly, calcineurin exhibits a high affinity toward the LxVP motif of NFATc1, c3, and c4, but a weak binding strength for NFATc2. Unlike in the immune cells with dominant expression of NFATc2, NFATc3 in glioma cells could still interact with calcineurin through an alternative binding site and remain activated despite blocking of the PxIxIT docking site by 11R-VIVIT. An engagement of this additional binding site in calcineurin-NFAT interaction may also explain the observed differences in the responses of glioma cells to 11R-VIVIT peptide and VIVIT-GFP. GFP fused to VIVIT may be a spherical obstacle, blocking the binding of calcineurin to LxVP and leading to decreased activation of NFAT. Recently, Wang et al. [23] designed a bioactive peptide against two sites of calcineurin/NFAT interaction, targeting both PxIxIT and LxVP motifs. It would be worth checking whether a blockade of both docking sites might result in an inhibition of NFAT activity in glioma cells. However, a synthetic LxVP peptide blocks not only calcineurin–NFAT interaction, but also calcineurin phosphatase activity [22], which compromises a peptide specificity.

Polyarginine oligopeptides have been developed to deliver bioactive peptides and proteins into eukaryotic cells and have been shown to exhibit greater efficiency than other CPPs [24]. Yet, efficient entry of nona-arginine (9R) cell-penetrating peptides into adherent cells at 10–20 µM has been recently linked with a transient increase in intracellular calcium [25]. NFAT activity depends on calcium-regulated phosphatase calcineurin. The observed upregulation of NFAT signaling following administration of 11R-VIVIT, a peptide with an oligoarginine CPP, might result from increased calcium levels. However, a lack of such a response of glioma cells treated with 11R-VEET, which contain the same CPP sequence and the same set of amino acid residues (just in a scrambled order), excludes this possibility. Noguchi et al. [26] reported that 11R-VIVIT at concentrations > 10 μM affects the viability of βTC6 cells (a β-cell line) in a manner dependent on the VIVIT sequence, not the 11R sequence. Based on the minimal sequence of the regulators of calcineurin (RCAN), responsible for the inhibition of calcineurin-NFAT signaling, they developed the RCAN1-11R peptide and showed that RCAN1-11R binds to calcineurin with high affinity and selectively interferes with the calcineurin/NFAT interaction without affecting cell viability [26]. The use of such peptide inhibitors could be an alternative for 11R-VIVIT in glioma cells. 

## 4. Materials and Methods

### 4.1. Materials

VIVIT peptide (MAGPHPVIVITGPHEE), Sim-2-VIVIT (AKAARQAARG-MAGPHPVIVITGPHEE), Sim-2-VEET (AKAARQAARG-MAGPPHIVEETGPHVI), 11R-VIVIT peptide (RRRRRRRRRRR-GGG-MAGPHPVIVITGPHEE), 11R-VEET (RRRRRRRRRRR-GGG-MAGPPHIVEETGPHVI) were synthetized by GeneScript. The peptides were purified by HPLC with >95% purity and were provided as hydrochloric salts. Anti-NFATc1 and anti-NFATc2 antibodies were from Thermo Scientific, anti-NFATc3 was from Cell Signaling Technology (Beverly, MA, USA), and anti-Lamin B (C-5) antibodies were from Santa Cruz Biotechnology, anti-GAPDH was from EMD Millipore (Burlington, MA, USA). Lipofectamine 2000 was from Invitrogen (Carlsbad, CA, USA). Nitrocellulose membrane and enhanced chemiluminescence detection system (ECL) were from Amersham Pharmacia Biotech. All other reagents were purchased from Sigma Aldrich (Saint Louis, MO, USA).

### 4.2. Cell Culture and Treatments

The established human glioma cell lines: T98G and LN18 (derived from glioblastomas, WHO grade IV), LN229, U251MG and U87MG (derived from astrocytomas WHO grade III) were from ATCC. Patient-derived glioma cell cultures WG4 (WHO grade IV) were developed as previously described [27]. Cells were grown in Dulbecco’s Modified Eagle’s Medium supplemented with 10% FBS (fetal bovine serum, Gibco, Paisley, UK) and antibiotics (50 U/mL penicillin, 50 µg/mL streptomycin) under standard conditions. Normal human astrocytes (NHA, Lonza, Basel, Switzerland) were cultured in Clonetics™/Poietics™ media and reagents. All peptides were dissolved in UltraPure distilled water (Thermo Scientific, Waltham, MA, USA) and added to culture media at indicated concentrations. Cyclosporin A (CsA, Sandimmun, Novartis, Basel, Switzerland) was used at 5 µg/mL or 1 µg/mL in LN229 and T98 cells, respectively. Ionomycin and phorbol 12-myristate 13-acetate (PMA) were used at the final concentrations of 1 μM and 50 nM, respectively. The effects of the treatments were monitored at various time points by phase-contrast microscopy. 

### 4.3. BrdU Incorporation Assay 

Cells were cultured in 96-well plates and treated with peptides for 48 h. BrdU (10 µM) was added to the culture medium for the last 2 h of treatment. Subsequently, the cells were fixed and the level of BrdU incorporation was determined according to the manufacturer’s protocol (Cell Proliferation ELISA BrdU assay, Roche Diagnostics GmbH, Mannheim, Germany). There were 5 biological replicates for each condition and at least 3 experiments.

### 4.4. Quantitative RT-PCR Analysis

Total RNA was isolated according to the manufacturer’s protocol (Promega, Madison, WI, USA), including a DNase digestion step. cDNAs were synthesized by extension of oligo(dT)_15_ primers using SuperScript III reverse transcriptase (Life Technologies, Carlsbad, CA, USA) in a mixture containing 1 μg of total RNA in 20 μL. Real-time qPCR analysis was carried out using the QuantStudio 12K Flex Real-Time PCR System (Applied Biosystems, Waltham, MA, USA) on cDNA equivalent to 10 ng RNA in 10 μL reaction volume. Human *NFATc1-c4* expression was measured using 1x TaqMan master mix (Thermo Fisher Scientific, Vilnius, Lithuania) and one of the following TaqMan gene expression assays (Life Technologies, Pleasanton, CA, USA): *NFATc1* (Hs00542678_m1), *NFATc2* (Hs00905451_m1), *NFATc3* (Hs00190046_m1) and normalized to *GAPDH* (Hs02758991_g1). Alternatively, the qPCR was run using 1x SYBR Green PCR master mix (Life Technologies, Carlsbad, CA) and 0.2 μM of each primer. The following primers were used: human *GAPDH* sense (5-‘ATCACCATCTTCCAGGAGCGA-3’) and antisense (5-‘AGCCTTCTCCATGGTGGTGAA-3’); *cMYC* sense (5-‘AAAACCAGCAGCCTCCCGCGA-3’) and *cMYC* antisense (5-‘AATACGGCTGCACCGAGTCGT-3’), *COX2/PTGS2* sense (5-‘AATCCTTGCTGTTCCCACCC-3’) and *COX2/PTGS2* antisense (5-‘AATTCCGGTGTTGAGCAGTTT-3’), and *CCND1* Quantitect Primer Assay (QT00495285, Qiagen, Hilden, Germany). Each pair of primers was validated for equal amplification efficiency to primers of the endogenous reference (*GAPDH*) at a wide range of cDNA concentrations. The specificity of the PCR reaction was confirmed by a single peak in the dissociation curve. Ct, the threshold cycle, was determined after setting the threshold in the linear amplification phase of the PCR reaction and averaged for each sample assayed in duplicates. ∆Ct for a particular gene was defined as Ct(target gene)-Ct(*GAPDH*). Data were analyzed with the Relative Quantification (^ΔΔ^Ct) method using QuantStudio 12K Flex software (Life Technologies, Carlsbad, CA, USA).

### 4.5. Immunocytochemistry 

Cells grown on glass coverslips were fixed in 4% paraformaldehyde for 10 min at room temperature, permeabilized with 100% MetOH for 10 min at −20 °C and incubated in a blocking solution containing 3% donkey serum and 10% FBS in 0.1% Triton-X-100 in PBS. Immunostaining was performed with mouse anti-human NFATc1 and rabbit anti-human NFATc3 antibodies, followed by donkey anti-mouse or anti-rabbit AlexaFluor 555-conjugated antibodies, respectively. After final washing in PBS, the coverslips were dried, mounted on slides with Vectashield Vibrance antifade mounting medium with DAPI (Vector Laboratories, Burlingame, CA, USA) and visualized with a confocal microscope (Fluoview FV10i, Olympus Corp., Tokyo, Japan).

### 4.6. Transfection and Luciferase Reporter Assay 

The expression plasmids (0.5 µg per 10^5^ cells) were delivered to the cells by electroporation using Amaxa™ 4D–Nucleofector™ system (Lonza, Cologne, Germany) in SE nucleofection buffer according to the manufacturer’s protocol. The cells were transfected with the GFP-VIVIT expression vector (Addgene, Watertown, MA, USA), which contained an oligonucleotide coding for MAGPHPVIVITGPHEE (VIVIT peptide) at the N-terminus of GFP or with the backbone control pEGFP-N1 plasmid (Addgene, Watertown, MA, USA). 

For evaluation of the NFAT transcriptional activity, the cells were seeded at 2.5 × 10^4^ cells/well in 48 wells plates and after two days transfected with pGL3-NFAT luciferase reporter plasmid (0.3 µg/well, Addgene) using Lipofectamine 2000 reagent (Invitrogen). The reporter plasmid contained three copies of the NFAT site from the minimal IL-2 promoter (−89 to +51). In the case of testing the effect of plasmid-delivered VIVIT, the reporter plasmid was introduced 24 h after the GFP-VIVIT expression vector. The exogenous peptides were added to the cells 1 h before the transfection. In some experiments, the cells were stimulated with 1 μM Io and 50 nM PMA or treated with CsA (1 µg/mL to T98 and 5 µg/mL to LN229 cell cultures) starting at 6 h post-transfection. Twenty-four hours after transfection the cells were lysed in 50 µL of a passive lysis buffer (Promega, Madison, WI, USA) and the luciferase activity was measured using the Luciferase Reporter System (Promega, Madison, WI, USA). Luciferase activity in cell lysates was normalized to the amount of protein, determined by the BCA Protein Assay Kit (Pierce, Waltham, MA, USA). Cells were transfected in duplicates in each of at least 3 independent experiments.

### 4.7. Preparation of Protein Extracts and Western Blot Analysis

Whole-cell protein lysates were prepared by scraping cells into a lysis buffer containing phosphatase and protease inhibitors as previously described [28]. To obtain nuclear and cytosolic fractions cells were collected in ice-cold PBS, centrifuged for 5 min at 300× *g* and lysed in a buffer containing 10 mM PIPES [pH 6.8], 100 mM NaCl, 1.5 mM MgCl2, 300 mM sucrose, 0.5% Triton X-100, 1 mM DTT and Complete Protease Inhibitor Cocktail (Roche) for 20 min on ice. Lysed cells were then centrifuged for 5 min at 1000× *g* to obtain the soluble (cytoplasmic) fraction. The pellet (nuclear fraction) was washed once, re-suspended in the same buffer and sonicated briefly. Protein extracts were mixed with 4 × Laemmli Sample Buffer, then boiled for 5 min, cleared by centrifugation for 15 min at 14,000× *g* and resolved on SDS-PAGE before electrophoretic transfer onto a nitrocellulose membrane. After blocking in 5% low-fat milk in TBS-T (0.1% Tween 20/Tris-buffered saline, pH 7.6) the membranes were incubated overnight with primary antibodies diluted in the blocking buffer and then for 1 h with relevant horseradish-conjugated secondary antibodies. Immunocomplexes were detected using an enhanced chemiluminescence detection system ECL (Amersham, Germany) either by membrane exposure to X-ray film or visualization in ChemiDoc Imaging System (Bio-Rad Laboratories, Hercules, CA, USA). The molecular weight of proteins was estimated with pre-stained protein markers (Thermo Fisher Scientific, Vilnius, Lithuania). Densitometric analysis was performed using NIH ImageJ 1.53 e software.

### 4.8. Statistical Analysis

All quantitative data are presented as mean ± SD. Statistical analyses were performed using Student’s *t*-test or, for multiple comparisons, with one-way ANOVA followed by post hoc Tukey or Dunnett tests using the GraphPad Prism software (GraphPad, Inc., San Diego, CA, USA). The *p*-values < 0.05 were considered to be statistically significant.

## 5. Conclusions

The development of short peptides disrupting protein–protein binding at functional sites is an excellent strategy for the exploration and modulation of protein functions through controlled interference with protein–protein interactions. The selection of a CPP for conjugation to the interfering peptide in order to facilitate its translocation into the cell should be adjusted to the target cell type. Sim-2 was not effective in translocating the VIVIT peptide into glioma cells but we reported the activity of 11R-VIVIT. Our results suggest that the VIVIT peptide produced inside the cells as a fusion protein with GFP prevented the binding of NFAT to calcineurin and its subsequent dephosphorylation, which led to the decreased activity of exogenous NFAT-responsive promoter and reduced transcription of endogenous NFAT target genes. These results were not reproduced with synthetic peptides and unexpectedly the VIVIT peptide activated NFAT translocation and NFAT-driven transcription. These effects may result from the cell-specific expression of NFAT transcription factors, mainly NFATc3 in glioma cells and mainly NFATc2 and NFATc1 in immune cells, in which the peptides were initially tested. NFATc3 activation engages two calcineurin docking sites, i.e., PxIxIT, which is targeted by 11R-VIVIT, and LxVP, in contrast to PxIxIT motif only in NFATc2. A spherical obstacle generated by GFP might explain the higher efficacy of VIVIT-GFP fusion protein in glioma cells as compared to 11R-VIVIT. In summary, we conclude that none of the tested synthetic VIVIT peptides is currently suitable for the evaluation of NFAT targets in glioma cells. The VIVIT peptide without a CPP or with Sim2-VIVIT did not show any activity in these cells and 11R-VIVIT lacked its inhibitory function on NFAT signaling.

## Figures and Tables

**Figure 1 molecules-26-04785-f001:**
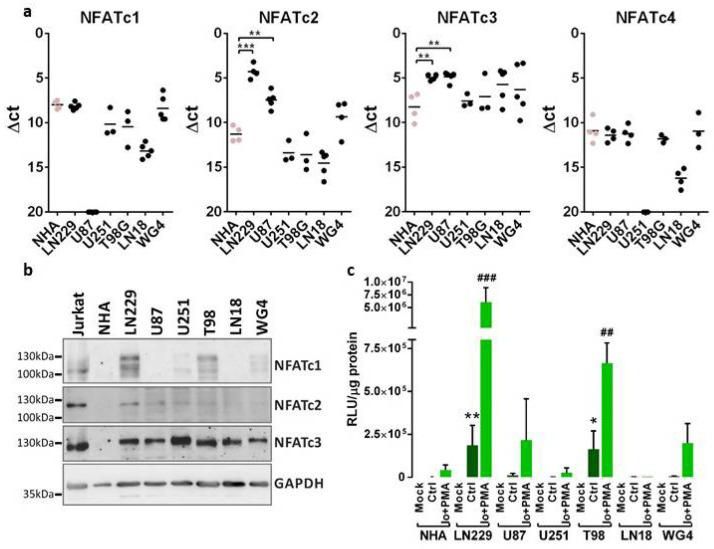
NFAT expression and transcriptional activity across glioma cell lines. (**a**) The expression of *NFAT mRNA* was analyzed by qPCR using TaqMan expression assays in established and primary (WG4) glioma cell cultures and in normal human astrocytes (NHA). Results are presented as ∆Ct values (Ct of a target gene—Ct of a reference *GAPDH* gene). Individual samples from at least 3 independent cell passages are plotted and a mean in each group is marked with a horizontal line. Statistical significance of changes vs. NHA was determined using one-way ANOVA followed by post hoc Dunnett test for multiple comparisons. (**b**) Whole-cell extracts were resolved by SDS-PAGE and subjected to immunoblot analysis using Abs against NFATc1, NFATc2, NFATc3 and GAPDH. Cellular extracts from human acute T-line lymphoblastic leukemia Jurkat cells served as a positive control in the determination of NFAT protein levels. (**c**) Glioma cells were transfected with a construct carrying a gene encoding luciferase under the NFAT-responsive promoter. The cells were left untreated or were stimulated with 1 µM ionomycin (Io), a calcium ionophore and 50 nM phorbol-12-myristate-13-acetate (PMA), an activator of protein kinase C. Luciferase activity was analyzed 24 h post-transfection. Bars represent means ± SD from at least 3 experiments carried out in duplicates. Statistical significance of changes vs. NHA was determined using *t*-test. Statistical significance is indicated as follows: * *p* < 0.05, ** *p* < 0.01, *** *p* < 0.001 vs. NHA Ctrl or ^##^
*p* < 0.01, ^###^ *p* < 0.001 vs. NHA Io + PMA.

**Figure 2 molecules-26-04785-f002:**
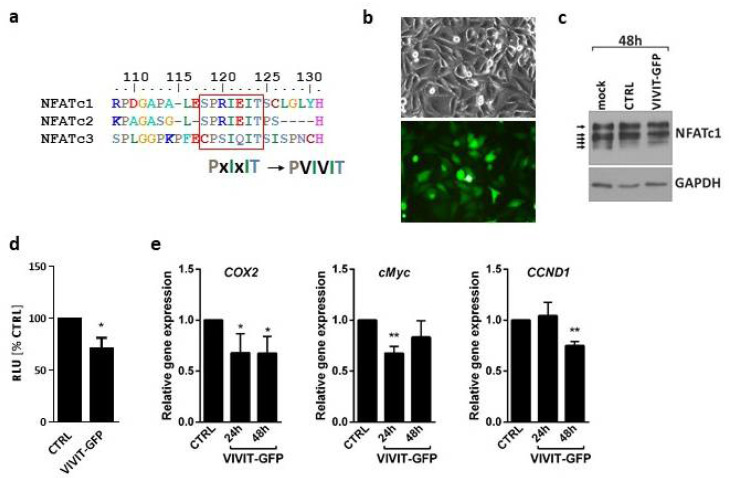
Overexpression of VIVIT-GFP blocks NFAT activity in glioma cells. (**a**) Sequences of calcineurin docking site in NFATc1, NFATc2 and NFATc3 and VIVIT. (**b**) Microphotographs of LN229 cells 48 h after transfection with VIVIT-GFP construct in phase-contrast and fluorescence microscope (upper/lower panel, respectively). (**c**) Western blot analysis of NFATc1 levels in mock transfected LN229 cells and in cells transfected with a control vector (CTRL) or VIVIT-GFP. GAPDH detection was used as a loading control. (**d**,**e**) Changes in the transcriptional activity of NFAT factors in VIVIT-GFP-transfected cells. Transcriptional activity of NFAT proteins was analyzed by luciferase reporter assay using pGL3-3xNFAT-luc plasmid (**d**) or by qPCR (**e**) to measure mRNA levels of endogenous NFAT target genes at indicated time points. Statistical significance of changes vs. CTRL was determined using one sample *t*-test (* *p* < 0.05, ** *p* < 0.01 ).

**Figure 3 molecules-26-04785-f003:**
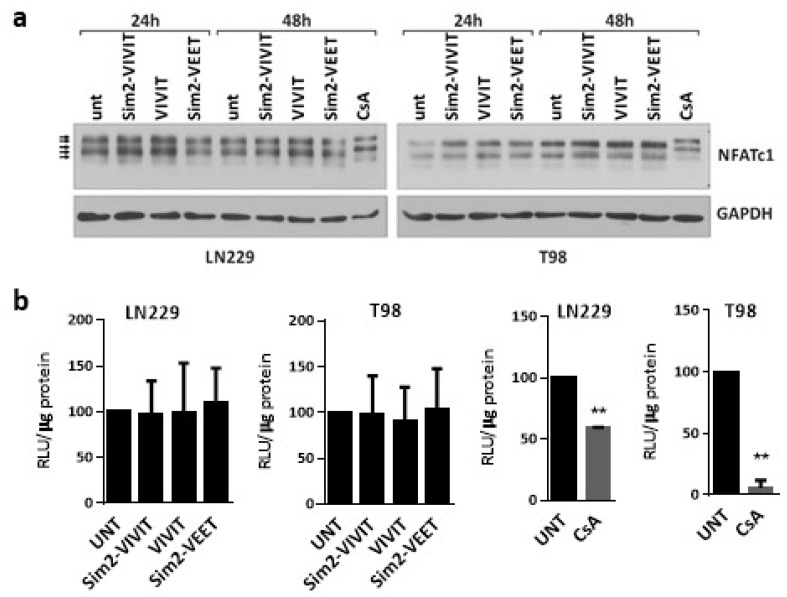
Sim2-VIVIT does not interfere with NFAT signaling in glioma cells. (**a**) Western blot analysis of NFATc1 levels in LN229 cells treated for 24 h and 48 h with the peptides at the final concentration of 500 µM. GAPDH detection was used as a loading control. (**b**) Measurement of NFAT-dependent promoter activity using luciferase reporter assay in LN229 and T98 cells transfected with pGL3-3xNFAT-luc plasmid 6 h prior to treatments with 500 µM of each peptide or with CsA (at 5 µg/mL in LN229 and 1 µg/mL in T98 cells) for 18 h following the transfection. Statistical significance of changes vs. untreated cells (UNT) was determined using one sample *t*-test (** *p* < 0.01).

**Figure 4 molecules-26-04785-f004:**
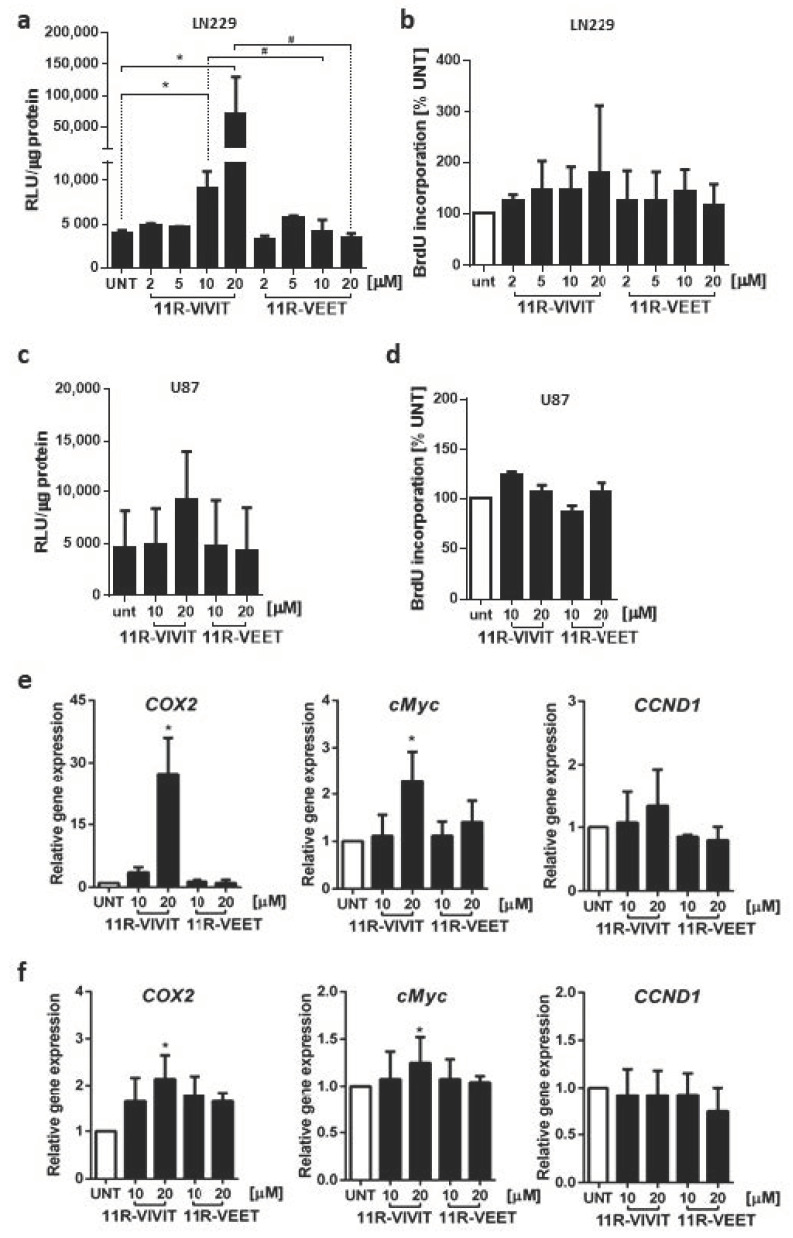
The effect of 11R-VIVIT on NFAT activity and proliferation of glioma cells. NFAT-dependent promoter activity was measured using luciferase reporter assay in LN229 (**a**) and U87 (**c**) cells transfected with pGL3-3xNFAT-luc plasmid and treated for 25 h with the peptide at indicated concentrations (including 1 h pre-treatment prior to transfection). Bars represent means ± SD from at least 3 experiments carried out in duplicates. Statistical significance of changes vs. untreated cells (UNT) or 11R-VEET-treated cells was determined using one-way ANOVA followed by Tukey post-hoc test for multiple comparisons (* *p* < 0.05 vs. UNT, # *p* < 0.05 vs. 11R-VEET). The proliferation of LN229 (**b**) and U87 (**d**) cells was evaluated after 48 h treatment with the peptides at the indicated concentrations. mRNA levels of endogenous NFAT target genes were analyzed by qPCR in LN229 (**e**) and U87 (**f**) cells after 6 h of treatment. Statistical significance of changes vs. untreated cells (UNT) was determined using one sample *t*-test (* *p* < 0.05).

**Figure 5 molecules-26-04785-f005:**
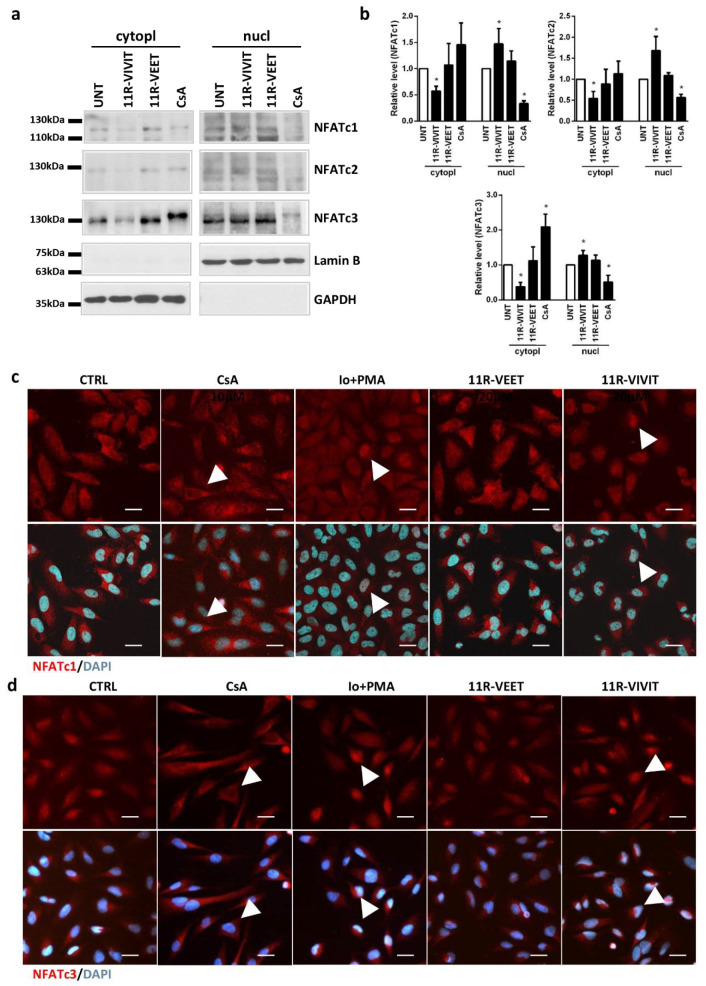
11R-VIVIT enhances NFAT translocation to the nucleus. (**a**) Distribution of NFATc1–c3 in cytosolic and nuclear fractions was evaluated using Western blotting 4 h after treatment of LN229 cells with 11R-VIVIT (20 µM), 11R-VEET (20 µM) or CsA (5 µg/mL). (**b**) The graphs show the relative intensity of the bands on immunoblots as compared to the control (UNT). The levels were based on the densitometric evaluation and normalized to GAPDH or Lamin B levels for cytosolic or nuclear fractions, respectively (the mean +/− SD of three independent experiments). Statistical significance of changes was determined using one-sample *t*-test (* *p* < 0.05), (**c**,**d**) Subcellular localization of NFATc1 (**c**) and NFATc3 (**d**) in LN229 cells 4 h after treatment with 11R-VIVIT, 11R-VEET (both at 20 µM), 1 μM Io/50 nM PMA or CsA (5 µg/mL). A scale bar corresponds to 25 µm.

## Data Availability

The data presented in this study are available on request from the corresponding authors.

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
