# Peer review of "Delivery of the VIVIT Peptide to Human Glioma Cells to Interfere with Calcineurin-NFAT Signaling"

_molecules, 2021, doi:10.3390/molecules26164785_

Round 1
Reviewer 1 Report
In this manuscript, the authors investigated the NFAT-inhibitory peptide VIVIT in human glioma cell line models. The authors assessed the expression of NFATc1, NFATc2, NFATc3 and NFATc4 in human glioma cell lines on messenger RNA and protein level and used a luciferase-reporter assay under the control of a NFAT-responsive promoter. Plasmid-based overexpression of VIVIT decreased NFAT activity. In contrast, cell-penetrating forms of VIVIT (Sim2-VIVIT and 11R-VIVIT) lacked NFAT-inhibitory function. Unexpectidly 11R-VIVIT even led to a nuclear translocation of NFAT and an increase in NFAT-associated transcriptional activity.
This is a report of overall negative results, which are still important for sharing. In the discussion the authors speculate about an alternative NFAT-calcineurin docking site that is not inhibited by the VIVIT peptide.
Major
For the western blot data in figure 5A, it would have been accurate to include also the control peptide 11R-VEET to confirm the immunofluorescence results form 5B also with a second method and in a more quantitative manner.
Minor
Figure 5B and C, the scale bar is missing and should be added
Author Response
We thank the reviewer for appreciation of our work and helpful suggestions regarding the presentation the results. The data we presented are not overall negative results, as firstly we report an extensive results on expression and activities of distinct NFAT proteins in several human glioma cells lines. Secondly, we demonstrate that the genetically delivered VIVIT-GFP is capable of inhibiting NFAT signalling in glioma cells. Indeed, CPP-VIVIT peptides were not effective in blocking NFAT signalling in glioma cells, which is disappointing but presenting such results has a value for researchers and points out to the use of other tools. The stimulatory activity of 11R-VIVIT was unexpected but we present all controls to show that this is not an artifact but an off target activity which should be taken into consideration in data analysis. We provide potential explanation for this unanticipated effect.
According to the reviewer's suggestion, we prepared an updated Figure 5a using new Western blot data that includes the treatment with a control peptide 11R-VEET and confirms the immunofluorescence results. Additionally, we provided the results of the densitometric analysis of immunoblots from 3 independent experiments (Figure 5b) to present the data in a more quantitative manner. We revised the manuscript accordingly.
We also added scale bars to Figure 5b and c.
Reviewer 2 Report
This work design a peptide VIVIT to bind to NFAT. It makes reader confused why Sim2-VIVIT and 11R-VIVIT did not show any anti-cancer action. Authors shoud solve this problem and restore the function of the VIVIT peptide.
Author Response
In our hands overexpression of the VIVIT peptide fused with GFP decreased the NFAT-driven activity and inhibited the transcription of endogenous NFAT-target genes in LN229 cells. Unexpectedly these results were not reproduced when the peptide was delivered extracellularly using Sim2 or 11R cell-penetrating peptides. Sim2-VIVIT did not show any effects even at a very high concentration of 500 µM, which is already beyond a pharmacologically reasonable range. It shows that Sim2 is not an effective CPP in glioma cells and as potential reasons why it does not work are numerous, it is beyond this manuscript.
With the 11R-VIVIT the situation is different as it definitely works intracellularly and its action is VIVIT dependent. We demonstrate that 11R-VIVIT led to a nuclear translocation of NFAT and an increase in NFAT-associated transcriptional activity. In the discussion of the manuscript we provided several plausible explanations of these results. Our interpretation is that it is due to potential interactions with the other binding site for the specific NFAT protein expressed in glioma cells. 11R-VIVIT showed cytotoxicity in LN229 cells at >50 µM but it was not related to NFAT inhibition.
We believe that publishing such results, even if they are different from what could be expected, is useful for researchers and provides additional information about the action of VIVIT and both CPP peptides.
Round 2
Reviewer 1 Report
The authors addressed all points sufficiently.